# GPCR Sense Communication Among Interaction Nematodes with Other Organisms

**DOI:** 10.3390/ijms26062822

**Published:** 2025-03-20

**Authors:** Jie Wang, Changying Guo, Xiaoli Wei, Xiaojian Pu, Yuanyuan Zhao, Chengti Xu, Wei Wang

**Affiliations:** 1Academy of Animal Science and Veterinary, Qinghai University, Xining 810016, China; wangjie08142023@163.com (J.W.); gchangying2021@163.com (C.G.); 18894499178@163.com (X.W.); pu_x_j@163.com (X.P.); 18893147262@163.com (Y.Z.); 2Key Laboratory of Northwest Cultivated Land Conservation and Marginal Land Improvement Enterprises, Ministry of Agriculture and Rural Affairs, Delingha 817000, China

**Keywords:** GPCRs, FLPs, pheromones, nematodes

## Abstract

Interactions between species give rise to chemical pathways of communication that regulate the interactions of transboundary species. The communication between nematodes and other species primarily occurs through the regulation of chemicals, with key species including plants, insects, bacteria, and nematode-trapping fungi that are closely associated with nematodes. G protein-coupled receptors (GPCRs) play a crucial role in interspecies communication. Certain *flp* genes, which function as GPCRs, exert varying degrees of influence on how nematodes interact with other species. These receptors facilitate the transmission of corresponding signals, thereby completing the interactions between species. This paper introduces the interactions between nematodes and other species and discusses the role of GPCRs in these organisms, contributing to a deeper understanding of the impact and significance of GPCRs in cross-border regulation between nematodes and other species. Furthermore, it is essential to leverage GPCRs in efforts to control pests.

## 1. Introduction

The communication system regulates a diverse array of behaviors through a conservative signaling network that involves small molecules, such as environmental signals and neurotransmitters, which arise from interactions among species [1]. This communication triggers chemical sensing, which is crucial for the survival of individuals and species. It also assists animals in locating nutritious food, selecting appropriate mating partners, and avoiding predators. Furthermore, it enables plants to initiate corresponding immune responses, thereby protecting them from damage caused by external pathogens and microorganisms [2,3]. Nematodes employ a diverse array of receptors and ion channels to detect small molecular substances and transmit signals both intracellularly and extracellularly. Among these, G protein-coupled receptors (GPCRs) play a pivotal role in this signaling process [4]. Consequently, GPCRs are essential for regulating various behaviors in nematodes.

GPCRs function as cell surface receptors situated in the plasma membrane, facilitating the transmission of external signals to the interior of cells [5]. GPCRs, recognized as one of the largest and most extensively studied protein families, play a pivotal role across various disciplines [6]. For example, novel pharmaceuticals have been developed by harnessing the functional properties of GPCRs in the medical sector. The model organism *Caenorhabditis elegans* had also been investigated in biological research, particularly concerning GPCRs. Furthermore, GPCRs contribute to biological control strategies for pest management and nematode management in agricultural practices [1,7,8,9]. In the twentieth century, Robert Lefkowitz and Brian Kobilka revolutionized our understanding of cellular signaling through their groundbreaking discovery of GPCR transduction mechanisms. Their contributions were recognized with the Nobel Prize in chemistry in 2012. To this day, GPCRs remain a focal point in the pharmaceutical industry, constituting approximately half of all available drugs [10]. Based on DNA sequencing, it is estimated that there are approximately 800 GPCRs in the human genome [11]. The GPCR family is primarily composed of a polypeptide chain featuring seven transmembrane (7TM) α-helices. Additionally, six rings connect the intracellular C-terminal domain to the extracellular N-terminal domain within this polypeptide chain [2,7,12]. The amino acid sequence of GPCRs is highly conserved, typically consisting of 200-1000 amino acids in length, and their three-dimensional (3D) structure is also remarkably conserved [9]. In the amino acid sequence of GPCRs, the key residues such as the DRY motif, NPxxY motif, and CWxP motif play a central role in signal transduction. The conservatism and functional diversity of these residues enable GPCRs to respond to a variety of ligands and activate downstream signaling pathways. The DRY motif, located at the intracellular end of transmembrane helix 3 (TM3), consists of Aspartic Acid-Arginine-Tyrosine (D-R-Y) and is closely related to G protein activation. Aspartic Acid (D) and Arginine (R) contribute to stabilizing the receptor conformation, while Tyrosine (Y) is involved in signal transduction [13]. The NPxxY motif, found at the intracellular end of transmembrane helix 7 (TM7), has a sequence of Asparagine-Proline-X-X-Tyrosine (N-P-x-x-Y) (x: any amino acid) and is primarily involved in receptor activation and G protein coupling. Tyrosine (Y) plays a significant role in signal transduction [13]. The CWxP motif, located in transmembrane helix 6 (TM6), consists of Cysteine-Tryptophan-X-Proline (C-W-x-P) (x: any amino acid) and is primarily responsible for maintaining receptor conformation. Notably, Tryptophan (W) and Proline (P) are crucial for the helical structure [14,15]. The GPCR superfamily consists of approximately 800 distinct receptor genes, of which around 400 are classified as nonolfactory receptors [12]. In biological systems, GPCRs are involved in numerous signaling pathways, regulating processes such as hormone regulation, neurotransmission, and sensory perception [16,17].

Pheromones represent a primary category of chemical signals that influence physiology, development, and inter-species regulatory behavior [18]. GPCRs, which function as pheromone receptors, play a crucial role in chemical communication between nematodes and various organisms, including fungi, bacteria, plants, and insects. The informational regulatory substance involved in this communication is ascaroside [9,19,20,21,22]. Additionally, other chemical substances, such as 2-heptanone and dimethyl trisulfide, also participate in inter-species interactions [23,24].

FMRFamide-like peptides (FLPs) represent the largest known family of neuropeptides in invertebrates. Recent research has identified 2,143 *flp* genes across more than 100 species of nematodes [25]. The signaling systems involving FLPs are central to the neuromuscular functions of nematodes. [26]. Most FLPs are recognized for exerting their effects through binding to GPCRs. This binding initiates a series of intracellular signal transduction events, ultimately influencing cellular functions and eliciting physiological responses [27,28,29]. Bioinformatics analysis using BLAST reveals significant conservation of FLPs in nematodes. Many species exhibit complementary FLP sequences that are intricate and resemble those of the model nematode *C. elegans*, which expresses 31 FLPs and encodes over 70 different peptides [30,31]. Within the phylum nematoda, at least 32 *flp* genes have been categorized, rendering the FLP system in nematodes remarkably complex. The diversity of the nematode FLP complement is particularly well-mapped in *C. elegans*, where more than 70 FLPs have been predicted. FLPs are expressed in the majority of the 302 neurons in *C. elegans*, including interneurons, sensory neurons, and motor neurons. The extensive expression of FLPs highlights the diverse functional roles of nematode FLP signaling, which encompasses neuroendocrine and neuromodulatory effects on locomotion, reproduction, feeding, and behavior [32]. Research has shown that certain genes within the FLP family are classified as belonging to the GPCR category. Notably, genes such as *flp-1*, *flp-2*, *flp-3*, *flp-4*, *flp-5*, *flp-7*, *flp-10*, *flp-11*, *flp-14*, *flp-15*, and *flp-18* have been identified [30]. Investigations into FLPs indicate that the time course of signal transduction via the GPCR pathway, along with its associated second messenger pathways, is relatively slow. Consequently, the effects of FLPs on muscles and nerves, mediated by GPCRs, correspond with the more traditional neuromodulatory effects attributed to neuropeptides [30]. Research on GPCRs has opened up a novel and highly promising avenue in the field of pest control. GPCRs play a central role in the physiological regulation processes of pests, including environmental perception, growth and development regulation, and behavior control. By thoroughly exploring the structure and function of GPCRs, we can accurately identify specific receptors that can serve as targets for control, enabling informed decision-making. However, there is a notable absence of a comprehensive summary detailing the specific FLP genes associated with GPCRs, as well as other GPCR receptor genes that play a role in regulating communication between nematodes and other species. This article aims to retrieve relevant studies on the interaction between nematodes and other species from databases such as PubMed, Web of Science, ScienceDirect, and the Wiley Online Library, and integrate and elaborate on the regulatory role of GPCRs in the interaction process between nematodes and other species, ultimately offering new research ideas and directions for future pest prevention and control.

## 2. Classification Systems of GPCRs Biology

Despite the presence of highly variable sequences, GPCRs exhibit similar conserved structures [10]. GPCRs comprise a large protein superfamily that is categorized into six classes: rhodopsin-like, secretin receptor family, metabotropic glutamate receptors, fungal mating pheromone receptors, cyclic AMP receptors, and frizzled receptors [10,33]. In humans, GPCRs are classified into four categories: class A (rhodopsin-like family), class B (secretin and adhesion), class C (glutamate), and class F (frizzled) [34].

The rhodopsin family of GPCRs is the largest among all vertebrates, comprising 715 members and accounting for 90% of all GPCRs in these organisms. Rhodopsin-like GPCRs exhibit a high degree of sequence similarity to the angiotensin receptor AT1 [35]. The ligands associated with this family include a diverse array of substances, such as small molecules (Angiotensin II) [36], neurotransmitters (Norepinephrine) [37], peptides (Angiotensin III and Angiotensin IV) [38,39], and hormones (Aldosterone) [40]. Notable rhodopsin-like GPCRs include the apelin receptor (APJ) [41], the adenosine A2B receptor (A2BAR) (belongs to the α-branch of rhodopsin-like GPCRs) [42], and gonadotropin-releasing hormone (GnRH) receptor, etc. [43]. The secretin GPCR family consists of 15 receptors in humans, with ligands that are relatively large polypeptide hormones, including glucagon, insulinotropic polypeptide, secretagogue, and growth hormone-releasing hormone [44]. Members of the secretin GPCR family are also found in invertebrates, such as Caenorhabditis elegans and Drosophila melanogaster. The secretin GPCRs exert broad physiological effects, with notable members including the glucagon receptor (GCGR) and the glucagon-like peptide 1 (GLP-1) receptor [45,46]. Additionally, metabotropic glutamate receptors (mGluRs) form a family of neuromodulatory GPCRs that regulate cell signaling and synaptic transmission related to the major excitatory neurotransmitter, L-glutamic acid [47]. Family C of GPCRs comprise eight metabotropic glutamate receptors, two heterodimeric gamma-aminobutyric acid receptors, a calcium-sensing receptor, three taste receptors, a promiscuous L-alpha-amino acid receptor, and five orphan receptors [48]. The fungal mating pheromone receptors, including the sterile α-factor receptors *Ste2* and *Ste3*, mediate polarized hyphal growth towards host-released peroxidase [49,50]. The vasopressin type 2 receptor, classified as a cyclic AMP receptor, undergoes a process of expression in which activation stimulates the accumulation of cyclic AMP (cAMP) [51]. As one of the most enigmatic GPCR families, it consists of ten frizzled (FZD1-10) and smoothened (SMO) [52] subtypes. This receptor primarily regulates the developmental Wnt signaling pathway [45,46]. Among various organisms, this family is one of the most highly conserved, being present in *C. elegans*, flies, and vertebrates [40]. However, the receptor sequences among GPCR classifications are limited, with each receptor classification exhibiting specific recognizable characteristics [53].

## 3. Functions and Features of GPCRs

GPCRs are predominantly made up of G proteins and the receptors themselves. The receptor component is a monomeric protein that typically consists of around 300 to 400 amino acid residues. In terms of structure, it possesses an extracellular N-terminal region that contains between 30 and 50 amino acids. Additionally, the peptide chain is distinguished by a configuration of seven alpha-helical transmembrane segments. Each of these hydrophobic transmembrane segments comprises approximately 20 to 25 amino acids, contributing to the overall structural integrity and functionality of the receptor [54,55,56]. The guanosine triphosphate (GTP) binding regulatory protein, commonly referred to as the G protein, is located on the cytoplasmic side of the plasma membrane. G proteins vary in size, ranging from small signaling proteins such as Ras to large heterotrimeric G protein complexes composed of α, β, and γ subunits [32,57,58]. The G protein, which interacts with both GTP and guanosine diphosphate (GDP), plays a central role in GPCR signaling, mediating a broad spectrum of physiological responses [28,57]. The α and γ subunits are tethered to the cell membrane through the attachment of covalently bound fatty acid chains. This structural configuration plays a crucial role in the functioning of G proteins, which act as molecular switches within the broader framework of signal transduction. In the resting state, the α subunit is bound to guanosine diphosphate (GDP), placing it in a conformation that can be described as closed. However, when this subunit encounters guanosine triphosphate (GTP), it undergoes a significant conformational change, transitioning into an open state that is essential for activating downstream signaling pathways. Furthermore, the α subunit possesses intrinsic GTPase activity, which enables it to facilitate the hydrolysis of the bound GTP, ultimately leading to the reversion of the G protein trimer to its inactive state. This cycle of activation and inactivation is fundamental to proper cellular signaling. Recent studies have highlighted the role of regulators of G protein signaling (RGS), which engage with GPCRs that are occupied by ligands. These interactions give rise to molecular complexes that exert opposing effects on G protein activity: on the one hand, RGS proteins exhibit guanine nucleotide exchange factor (GEF) activity that promotes the activation of G proteins, while on the other hand, they also possess GTPase-activating protein (GAP) activity, which serves to inactivate the G proteins. This dual functionality underscores the intricate regulation of G protein signaling pathways [57,59].

GPCRs are versatile molecular machines that regulate a wide range of physiological responses to diverse hormones and neurotransmitters [60]. Membrane proteins are essential for cell-to-cell communication, and GPCRs, as a subclass of membrane proteins, primarily detect a variety of extracellular signals, including photons, ions, nucleosides, lipids, peptides, and proteins [61]. They facilitate the transmission of chemical signals between cells by either activating or inhibiting G proteins [60]. The primary function of GPCRs is to transmit signals to cells. GPCR-associated receptor signaling serves at least four distinct roles: (1) the direct mediation of receptor signaling, (2) the regulation of receptor signaling, (3) linking receptors to various effectors, and (4) influencing other aspects of receptor pharmacology and function that can be modified [56,62].

## 4. G Protein-Coupled Receptor Signaling: Transducers and Regulation

Signal transduction facilitated by GPCRs can trigger a range of effects through different pathways. However, the essential pattern of signal transduction remains stable and includes the following key steps: (1) the binding of a ligand to the receptor; (2) the activation of the G protein by the receptor; (3) the activation or inhibition of downstream effector molecules by the G protein; (4) the alteration of second messenger content and distribution within cells by effector molecules; and (5) the interaction of second messengers with specific target molecules to modify their conformation, thus affecting metabolic processes, gene expression, and other cellular functions [28,56,63].

Cellular signaling pathways that are mediated by GPCRs predominantly include the cyclic adenosine monophosphate (cAMP) signaling pathway and the phosphatidylinositol signaling pathway. Within the framework of the cAMP signaling pathway, extracellular signals interact with specific receptors on the cell membrane. This interaction triggers a series of events that regulate the activity of adenylate cyclase, an enzyme responsible for catalyzing the conversion of ATP (Adenosine Triphosphate) into cAMP. As a result, the levels of cAMP, which functions as a crucial second messenger, increase, effectively transforming extracellular signals into corresponding intracellular signals. On the other hand, the phosphatidylinositol signaling pathway involves the engagement of extracellular signaling molecules with GPCRs found on the surface of the cell. This binding activates an enzyme known as phospholipase C (PLC-β) that is localized at the plasma membrane. The activation of PLC-β leads to the hydrolysis of phosphatidylinositol 4,5-bisphosphate (PIP2), a phospholipid also located at the plasma membrane. This hydrolysis releases two significant second messengers: inositol 1,4,5-triphosphate (IP3) and diacylglycerol (DG) [57]. Collectively, these processes illustrate the important mechanism by which extracellular signals are converted into intracellular responses, a phenomenon that is often termed a ‘second messenger system’ or a ‘double messenger system’.

## 5. Research Progress of GPCR Among Nematodes and Other Organisms

Nematodes are recognized as one of the oldest groups of species on our planet, and they can be classified into three primary categories based on their distinct lifestyles. These categories include free-living nematodes, which thrive independently in various environments; animal-parasitic nematodes, which rely on animal hosts for survival; and plant-parasitic nematodes, which infest and derive nutrients from plants. Among these, *C. elegans* has emerged as a particularly well-studied free-living nematode and is frequently utilized as a model organism in laboratory research settings. Its simplicity and well-mapped genetics make it an ideal candidate for exploring various biological processes. Currently, a substantial portion of research effort is directed toward understanding the interactions between *C. elegans* and other organisms, with a particular focus on the modifications and changes occurring in GPCRs. These receptors play critical roles in cellular communication and signal transduction, and thus, their dynamics in relation to *C. elegans* interaction with other species are of significant interest. Conversely, there remains a notable gap in the literature regarding the investigations into the relationships between parasitic nematodes and their respective hosts. This includes, but is not limited to, the interactions of entomopathogenic nematodes with insect hosts, plant-parasitic nematodes with various plant species, and the associations of nematodes with fungi. The forthcoming sections of this paper will delve into the advancements achieved in research concerning the induction of GPCRs, as influenced by the interactions among different nematode species and various other taxa. This exploration aims to shed light on the biochemical and ecological implications of these interactions and to enhance our understanding of the broader biological significance of nematodes within their ecosystems.

### 5.1. Responding for Interaction Between C. elegans and Related Bacteria

*C. elegans*, a free-living nematode found in soil and decaying organic matter, primarily feeds on bacteria but is vulnerable to pathogenic microorganisms [64]. Communication between *C. elegans* and bacteria encompasses two main aspects. Firstly, *C. elegans* can be infected by pathogenic microorganisms [65]. The natural pathogens and parasites that infect *C. elegans* were first described in 2008 and 2011 [66,67]. Secondly, *Escherichia coli* OP50, utilized as a food source in laboratory settings, plays a significant role in this interaction [64]. *C. elegans* possesses the ability to recognize various pathogens and initiate distinct immune responses accordingly [65,68]. The nervous system processes input signals from infected local sites, integrating them to coordinate immune response mechanisms. The microbial sensing mechanism primarily activates molecular immune pathways, enhancing resistance to pathogens by reducing their presence and eliminating infections. Research had shown that the mechanisms involved in the initial activation of defense pathways also regulate the immune system. Consequently, immune regulatory mechanisms are crucial for maintaining immune homeostasis, as excessive activation of immune pathways can be detrimental to the host. Nematodes utilize these mechanisms to adapt and respond to external pathogens [69,70]. While *E. coli* is the primary food source for *C. elegans*, food selection also constitutes an essential adaptive strategy for these nematodes. During food selection, chemical communication between *E. coli* and *C. elegans* enhances the expression of specific GPCR receptors.

*C. elegans* possesses a genome that encodes over 1000 GPCRs, and its heterotrimeric G protein repertoire consists of 21 Gα subunits, along with 2 Gβ and 2 Gγ proteins [71], Among these, the rhodopsin-like GPCR chemoreceptor family is one of the largest, with 244 potential expression sites identified. This discovery was made using classical GFP reporter gene technology [72]. Isabe et al. studied 68 orphan peptide GPCRs in *C. elegans* and identified 461 pairs of peptide GPCRs, along with additional ligands for the identified GPCRs. This research demonstrates a specific and complex combinatorial interaction between GPCRs and their ligands in *C. elegans*. In summary, a significant number of GPCRs have been identified in *C. elegans*, which play crucial roles in various biological processes, including cell communication and environmental responses [73].

In *C. elegans*, the neuronal GPCR NPR-8 (Figure 1) plays a critical role in modulating the organism’s defense against pathogens such as *Pseudomonas aeruginosa*, *Salmonella enterica*, and *Staphylococcus aureus*. NPR-8 achieves this by inhibiting the expression of epidermal collagen, with predominant expression observed in the AWB (Anterior White Bipolar neuron), AWC (Anterior Chemosensory neuron), and ASJ (Anterior Sensory Junction neuron) neurons. This receptor regulates the epidermal structure’s response to infection, potentially influencing the nematodes’ defense mechanisms through the control of genes associated with epidermal collagen [74]. The functional deletion of NPR-8 had been shown to enhance the resistance of *C. elegans* to pathogen infections [74], indicating that NPR-8 regulates related pathways that impact the survival of nematodes under pathogenic stress. In contrast, OCTR-1 (Figure 1), another neuronal G protein-coupled receptor, primarily inhibits the innate immune response of *C. elegans* in non-neural tissues when challenged by *P. aeruginosa* [75,76]. This regulatory mechanism involves OCTR-1 suppressing the expression of specific protein synthesis factors, including ribosomal protein RPS-1 and translation initiation factor EIF-3.J, thereby contributing to enhanced immunity. These findings imply that the OCTR-1 regulatory pathway may represent a conserved signaling mechanism employed by the nervous system to maintain protein homeostasis during host immune defense, with predominant expression in ASH (Anterior Sensory Head neuron) and ASI (Anterior Sensory Interneuron) neurons [75]. *P. aeruginosa* infection in *C. elegans* can induce dendritic degeneration. Notably, SRBC-48 protects the nematode from infection-related dendritic degeneration in an autonomous manner within AWC neurons. Nematodes lacking SRBC-48 are susceptible to pathogen infections during the early stages of life. This susceptibility not only results in dendritic degeneration but also shortens lifespan due to the uncontrolled activation of immune genes (DAF-16), thereby revealing the dual adverse effects of early infection on both nematode dendrites and lifespan [77]. Furthermore, studies have demonstrated that *C. elegans* lacking the neuronal GPCR NMUR-1 (Figure 1) exhibit varying survival rates against bacterial pathogens such as *Enterococcus faecalis* and *S. enterica*, suggesting that the *nmur-1* gene regulates distinct immune responses tailored to different pathogens [78]. NMUR-1 can discern various pathogens, enabling it to respond to distinct attacks through the engagement of different neural circuits. This process involves the regulation of C-type lectin expression and the inhibition of unfolded protein response genes. As a multifaceted environmental factor, food influences numerous physiological processes in organisms, including lifespan [79]. Concurrently, the *nmur-1*, *hsf-1*, and *daf-16* genes affect lifespan via various mechanisms, indicating that sensory factors impact lifespan through multiple pathways in the presence of abundant food [79]. Notably, the impairment of NMUR-1 function significantly influences the lifespan of *C. elegans* when fed different *E. coli* food sources, with this impact being contingent on the specific structure of *E. coli* lipopolysaccharides [79]. This suggests that NMUR-1 plays diverse roles in various aspects of *C. elegans* biology. The *Bacillus nematocida* B16 releases the volatile organic compound 2-heptanone, which can infect *C. elegans*. The perception of AWC neuron function and the *str-2* (Figure 1) gene-encoded GPCR is primarily mediated by 2-heptanone. This principal pathway is the PLC pathway, which is also associated with the activation of Gα subunits of *egl-30*/*gpa-3* [24]. The chemicals metabolized by bacteria possess the ability to modulate the physiological response of *C. elegans* to some extent. The study found that the innate immune deficiency caused by the mutation of the classical immune gene *pmk-1* (Figure 1) promotes the evasion behavior of *C. elegans*. When *C. elegans* encounters bacterial infection, innate immunity is compromised, and an alternative evasion defense strategy can be enhanced through the HECW-1/NPR-1 (Figure 1) module, suggesting that GPCRs in the neural circuit may receive inputs from the immune system and integrate both systems to better adapt to the defensive state [69,80]. In the process of food selection, *C. elegans* utilizes the ADL (Anterior Dorsal Labial neuron) neuron to regulate preference by inhibiting the function of SRH-220. This inhibition may influence nematode food selection via the peptide energy signals of FLP-4 and NLP-10, with the functions of FLP-4 or NLP-10 being modulated by SRH-220 (Figure 1) [81]. Furthermore, studies have shown that the neuropeptide system NLP-12 in *C. elegans* affects responses to food availability by intermittently regulating the activity of head and body wall motor neurons associated with the GPCRs CKR-1 and CKR-2 [82]. It had been demonstrated that the motor response to fluctuations in food availability is controlled by the conditional stimulation of NLP-12 in either the head or body wall motor circuit [82]. The response of *C. elegans* to the volatile chemical dimethyl trisulfide (DMTS), produced by *E. coli*, is concentration-dependent, affecting the attraction or avoidance behavior of nematodes. This behavioral shift is mediated by two distinct types of chemosensory neurons (AWC and ASH), both of which express the DMTS-sensitive seven-transmembrane protein-coupled receptor SRI-14 (Figure 1) [23]. However, AWC and ASH utilize different glutamate transmission mechanisms to relay signals to AIB (Amphid Interneuron B) interneurons. Ultimately, the activity of AIB is transformed into a concentration-dependent behavioral response to DMTS [23].

*C. elegans* interacts with pathogens and various microorganisms primarily through the exchange of chemical signals, which elicit neural responses mediated by GPCRs. Additionally, during the food selection process, bacteria can influence the lifespan of *C. elegans*. Broadly speaking, both processes involve the detection of external signals resulting from chemical interactions among species via their respective sensory neurons. These signals are subsequently transmitted to the corresponding receptors for recognition, leading to consequential behavioral and physiological modifications.

### 5.2. Communication Between Nematode-Trapping Fungi and Nematodes

Nematode-trapping fungi (NTF) are specialized microorganisms that prey on free-living nematodes using unique trapping structures. These fungi exhibit complex trapping mechanisms, including constricted rings and five distinct types of adhesive traps: sessile adhesive knobs, adhesive nets, adhesive posts, and non-constricted rings [83,84,85]. NTF play a crucial role in the biological control of plant-parasitic nematodes by preying on them when food sources become limited [86]. The predation process is activated upon the detection of conserved nematode pheromone ascaroside, leading to the formation of intricate trapping mechanisms [87]. Chemical inhibitors suggest that G proteins may play a role in the closure of constricted rings in another nematode-trapping fungus, *Drechslerella dactyloides* [88]. Research has demonstrated that G-protein signaling is essential for both ascaroside sensing and trap formation in *A. oligospora*. Furthermore, the receptors for ascarosides in *A. oligospora* are likely GPCRs [87]. *A. oligospora* can attract nematodes, a process facilitated by a set of olfactory neurons in *C. elegans*, potentially involving various GPCRs [89]. This suggests that GPCRs may be involved in initiating the nematode’s response to *A. oligospora*. In this study, Chen et al. demonstrated that the conserved cAMP-PKA signaling pathway is crucial for trap morphogenesis in *A. oligospora*, although the associated GPCR has yet to be identified [90].

The nematode-trapping fungus *A. oligospora* serves as a model species for studying nematode-trapping fungi and holds indispensable significance across multiple research domains. Studies have indicated that Gin3 may partially activate the pheromone-responsive MAPK signaling pathway, thereby inducing the expression of GIN1, GIN7, and GIN8. This may suggest their potential roles as nematode-responsive genes in *A. oligospora* and as-yet unidentified nematode signal receptors [91]. *C. elegans* is capable of secreting the pheromone ascaroside, which can induce *Arthrobotrys flagrans* to form traps. Studies have demonstrated that the three GPCRs of *A. flagrans* (GprC, GprB, and GprD) play significant roles in the communication process with *C. elegans* [92]. However, research on GPCRs related to the interactions between the nematode-trapping fungus of the genus Arthrobotrys and *C. elegans*, as well as other nematode species, remains limited. Further studies are necessary to elucidate the role of GPCRs in the signaling mechanisms of the nematode-trapping fungus of the genus Arthrobotrys, particularly through the application of chemicals secreted by nematodes to induce various changes in the nematode-trapping fungus of the genus Arthrobotrys, thereby facilitating in-depth investigation.

### 5.3. Communication Responses of Host Insects Infected by Entomopathogenic Nematodes

Entomopathogenic nematodes, as soil microorganisms that parasitize insects, have been produced on a large scale for integrated pest management in agriculture. They serve as biological agents to control several significant pests, primarily including *Steinernema* and *Heterorhabditis nematodes* [93,94]. Research has demonstrated that the FMRFamide-like peptide (*flp*) gene is up-regulated during the interaction of different entomopathogenic nematodes with host insects. This gene promotes the behavior of nematodes in seeking various hosts, alongside insulin-like peptides and neuropeptide-like proteins, such as those found in *Ancylostoma ceylanicum*, *Globodera pallida*, and *Brugia malayi* [95,96,97].

In *Steinernema* nematodes, the significant up-regulation of four srsx GPCR genes (Sc-srsx-25v, Sc-srsx-3ii, Sc-srsx-22i, and Sc-srsx-24ii) is associated with the increased chemotaxis of *S. carpocapsae* UK1 and *S. carpocapsae* All towards the insect host *G. mellonella* (Figure 2). In contrast, the Sc-srt-62 chemosensory GPCR gene was down-regulated in S. carpocapsae UK1. Similarly, the Sc-npr-23 gene of GPCRs was independently targeted by two differentially expressed microRNAs (Figure 2). These distinct expression patterns are linked to chemotactic behavior, suggesting that microRNAs may play a role in modulating behavioral changes [21]. Furthermore, several microRNAs have been predicted to participate in the regulation of GPCRs in entomopathogenic nematodes, including *ador-1*, *gar-2*, *npr-11*, *npr-23*, *npr-26*, C30A5.10, *srsx-24*, and *gar-1*. All of these genes are annotated based on the *C. elegans* genome, although their functions in entomopathogenic nematodes remain unclear [21]. The FMRFamide-like peptide (*flp*) gene has been shown to be synergistically up-regulated during the host-seeking process in various parasitic nematodes [97]. Specifically, *flp-21*, as a FMRFamide-like peptide, is involved in regulating the behaviors of *S. carpocapsae*, which depend on sensory perception and are related to host-finding (Figure 2) [96]. Genes *flp-3*, *flp-7*, *flp-14*, and *flp-18* are highly expressed at the infective juvenile stage and progressively decrease by 15 h post-infection (Figure 2) [98]. Hba_15737 (*npr-9*) is a neuropeptide GPCR implicated in foraging behavior in *Heterorhabditis bacteriophora* (Figure 2) [99,100]. Entomopathogenic nematodes have been shown to possess associated GPCRs identified through transcriptome sequencing technology, with their functions annotated based on related gene functions in *C. elegans*. This confirms the active involvement of GPCR family genes in the host insect-finding process, along with contributions from certain genes within the FLP family.

Entomopathogenic nematodes utilize relevant genes associated with GPCRs to communicate with their host by infecting host insects and releasing specific chemicals. GPCRs respond by transmitting signals through various expression pathways. Ultimately, the hosts succumb, providing essential nutrients for the growth and reproduction of entomopathogenic nematodes. These behaviors primarily involve the regulation of FLP family genes and other genes within entomopathogenic nematodes.

### 5.4. Chemical Responses of Plant Nematodes to Host Plants

Plant nematodes are widely distributed globally and cause significant economic losses to various plants [101]. These nematodes primarily parasitize plants during their feeding and developmental stages, which impedes the absorption of nutrients and water, resulting in poor plant growth and reduced yield [9]. The main plant nematodes are *Meloidogyne incognita* [9], *Globodera pallida* [102], and *M. graminicola* [103] etc. Recent discoveries indicate that GPCRs can regulate the migration of plant nematodes to their hosts and influence the infection process, as GPCRs serve as targets for a variety of pharmacological agents [9]. The primary GPCRs in plant nematodes are predominantly derived from the FMRFamide-like peptides (FLPs) family, which is categorized as FMRFamide-related peptides (FaRPs) and constitutes the largest neuropeptide family in nematodes. Research shows that *flp* expression is up-regulated in the infective larval stage of several nematode parasites [25]. Structural information regarding FLPs is largely sourced from *C. elegans*, while functional data is primarily derived from *Ascaris suum*, which serves as a physiological model for nematodes [104].

The research demonstrates that the corresponding tobacco strain is constructed using two genes, *flp-14* and *flp-18*, derived from the J2 stage of *M. incognita* (Figure 3). This construction results in a 50–80% reduction in the resistant infection and proliferation of seedlings to *M. incognita* [104]. The application of in vitro gene silencing and host-induced gene silencing has shown that Mi-flp1, Mi-flp12, and Mi-flp18 (Figure 3) can be effectively utilized together to control *M. incognita*, leading to a significant reduction in the nematode multiplication factor (MF) [105]. Notably, these three genes exhibit varying expression levels at different stages of *M. incognita*. The maximum expression levels are observed in infective J2, while the expression levels in J3, J4, and young females are down-regulated, with the lowest expression levels found in eggs. This structural understanding can be leveraged to control the population of *M. incognita* and prevent plant injury [105]. Furthermore, plant transformation via Agrobacterium-mediated rice with host-induced gene silencing of Mg-flp-1 and Mg-flp-12 (Figure 3) resulted in reductions of 31–50% and 34–51%, respectively, in the endoparasites associated with the Mg-flp-1 and Mg-flp-12 transgenes. Similarly, the number of eggs per plant in transgenic plants was significantly reduced compared to wild-type plants [103]. The *flp-32* gene (Figure 3) is widely expressed in plant nematodes. Research on the *flp-32* gene in *G. pallida* has shown that silencing *flp-32* increases the diffusion rate and enhances the ability to infect potato plant roots. Based on these findings, the *flp-32* gene can be more effectively employed to manage the infestation of plant nematodes in crops [102]. The discovery revealed that GPCRs and neuropeptide signaling pathway genes exhibit their highest expression in the second-stage juveniles (J2s) of *G. pallida*, suggesting a correlation between this expression and the dispersal and host-searching behavior of nematodes [106,107]. Silencing the *flp-18* GPCRs derived from *M. graminicola* J2s resulted in a significant decrease in the penetration of rice and wheat roots by the nematode, indicating that this could be effectively controlled [108]. In this section, the verification of plant nematode GPCRs was conducted through RNA interference (RNAi), revealing the involvement of FLP family genes in the regulatory process. Additionally, genes from other families were subjected to functional verification via RNAi, such as the Bxy-npr-21 gene of *Bursaphelenchus xylophilus* is a receptor gene implicated in sexual attraction. Bioinformatic analyses reveal that it encodes a GPCR. RNAi experiments demonstrate that it is primarily involved in behaviors such as nematode movement, feeding, and mating. This research establishes a crucial foundation for the further prevention and control of *B. xylophilus* [109].

In the interaction between plants and plant nematodes, certain receptor genes from the FLP family, which are related to GPCRs, such as *flp-1*, *flp-12*, *flp-14*, *flp-18*, and *flp-32*, play a crucial role in the infection of plant nematodes. Additionally, other receptor genes may also significantly contribute to the induction process. The utilization of these identified related genes can further facilitate the study of the induction mechanisms of GPCRs and explore their applications in agricultural production.

## 6. Outlook and Future Directions

The interactions between nematodes and other species are predominantly facilitated through chemical communication, wherein GPCRs serve as receptors that mediate these responses. GPCRs are essential in physiology, as they translate extracellular signals into downstream intracellular responses. Due to their critical role in cell signaling, GPCRs remain primary targets for contemporary drug development efforts [110,111]. Building upon the significance of GPCRs in drug research, a comparable approach has been utilized in the development of drugs aimed at targeting plant nematodes. Furthermore, genes associated with GPCRs involved in the interactions between nematodes and other species have been identified. Their primary function in neural induction is to regulate immune responses, feeding behavior, and the infection of other hosts.

Functional verification and high-throughput sequencing technology have confirmed that certain GPCRs play a crucial role in nematode immunity. The GPCRs primarily discussed in this review include *npr-8*, *octr-1*, *nmur-1*, *npr-1*, *srbc-48*, and *str-2* (Table 1). The immune response of *C. elegans* against pathogens is mediated by the regulation of these GPCRs, facilitating chemical communication between species. As the nematode species with the highest number of identified GPCRs, *C. elegans* provides a substantial theoretical foundation for the discovery of GPCRs in other nematodes and their interacting species. For instance, the FLP family gene identified in *C. elegans* had been implicated in the immunity of parasitic nematodes. Investigating GPCRs in the feeding behavior of *C. elegans* enhances our understanding of how food influences various growth and developmental stages, as well as the process of dauer induction. This research will further elucidate the molecular mechanisms underlying the interaction between *E. coli* and *C. elegans*. Consequently, the ongoing identification and characterization of novel GPCRs derived from *C. elegans* significantly contribute to a comprehensive understanding of related immune mechanisms and behaviors.

In the communication between various plants and plant nematodes, the latter can secrete related pheromones and chemical substances. Upon sensing these signals, plants can initiate self-related immune mechanisms, which reduce the infection caused by plant nematodes [107]. The GPCRs of these plant nematodes have been found to be instrumental in controlling nematode infections in plants, as well as in the host-seeking behavior of the nematodes. This is primarily due to the disruption of neuropeptides involved in GPCR signal transduction, which interferes with the behavior and migratory capabilities of plant nematodes [108]. This paper primarily discusses how different plant nematodes infect host plants and how species interactions result in varying expression patterns of GPCRs among different nematode species, specifically including the expressions of *flp-1*, *flp-12*, *flp-14*, *flp-18*, and *flp-32* (Table 1). Furthermore, there is a relative scarcity of research on the communication of GPCR receptors in the interactions between nematode-trapping fungi and nematodes. As is well known, certain chemical substances can induce nematode-trapping fungi to produce traps, which subsequently kill the nematodes. This phenomenon can be exploited to control plant nematodes (Figure 4). However, the receptor for the induction pheromone ascarosides in nematode-trapping fungi has yet to be defined and validated. Further studies are necessary to explore the receptors associated with the induction pheromone ascarosides of *C. elegans*. In comparison to the model nematode species *C. elegans*, research on GPCR genes in non-model nematode species remains limited, indicating a need for further investigation into the role of GPCRs in interspecies communication in the future.

Entomopathogenic nematodes enhance their ability to locate host insects by secreting specific chemical substances and pheromones, allowing them to kill their hosts in a relatively short time, thereby facilitating their growth and reproduction (Figure 4). However, studies validating the role of GPCRs (Table 1) in the interaction between entomopathogenic nematodes and their hosts are limited, indicating a need for further research in this area.

In summary, we can gain a deeper understanding of the molecular mechanisms and processes involved in cross-border species regulation by elucidating the chemical exchanges between different nematodes and their interacting species. These species primarily receive extracellular chemical signals and transmit them internally through GPCRs. As a family of receptor genes, GPCRs play a crucial role in biological behavior and physiological regulation. Ongoing research continues to enhance our knowledge, facilitating further exploration and discovery.

GPCRs hold significant promise for pest control. Research has demonstrated that through computational biology methods, cell screening, and in vivo toxicity testing, a series of Allatostatin Type-C Receptor activators suitable for specific insecticides targeting *Thaumetopoea pityocampa* have been identified. These novel ligands are specific to lepidopteran larvae while being harmless to coleopteran larvae and adults. They represent a significant advancement in the design of next-generation insecticides aimed at insect GPCRs, offering a new approach to mitigating the damage caused by *T. pityocampa* and in developing innovative insecticides [112]. The GPCR protein NPFF2, located on the cell surface of *Bemisia tabaci*, plays a role in the trade-off between resistance and reproductive cost. Consequently, some studies have utilized GPCR inhibitors and the GPCR gene NPFF2 in managing the resistance of *B. tabaci* to neonicotinoid pesticides [113]. However, the translation of GPCR research into practical pest control methods faces numerous limitations and challenges. The complexity of physiological functions and the interplay of signaling pathways may result in unpredictable side effects when implementing control strategies that target GPCRs. Additionally, uncontrollable field factors and the development of pest resistance can diminish the efficacy of the developed compounds. Therefore, future efforts must focus on overcoming the barriers to translating GPCR research into field-applicable pest control methods by enhancing basic GPCR research, fostering ligand innovation, and optimizing field application technologies.

## Figures and Tables

**Figure 1 ijms-26-02822-f001:**
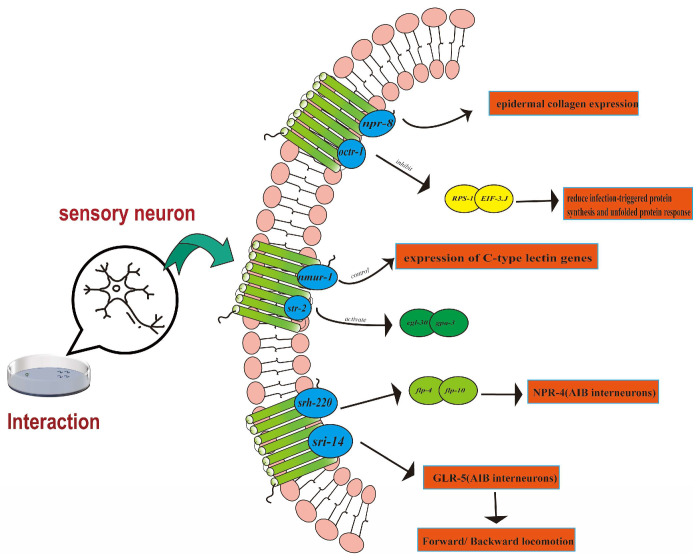
Related GPCR of bacterial interaction with *C. elegans*.

**Figure 2 ijms-26-02822-f002:**
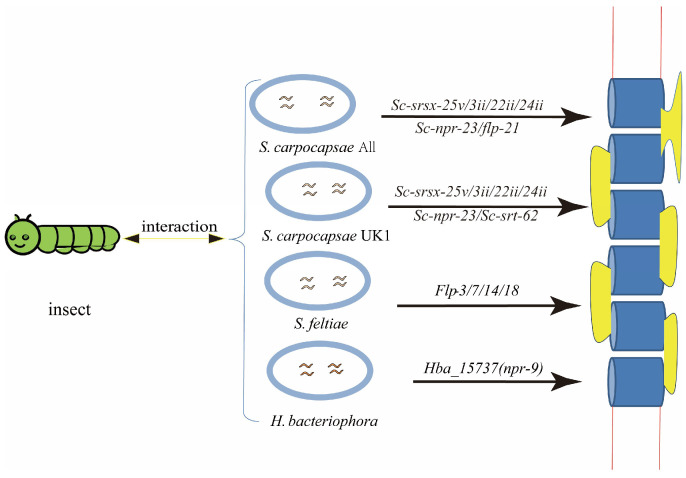
Related GPCR of insect with entomopathogenic nematodes.

**Figure 3 ijms-26-02822-f003:**
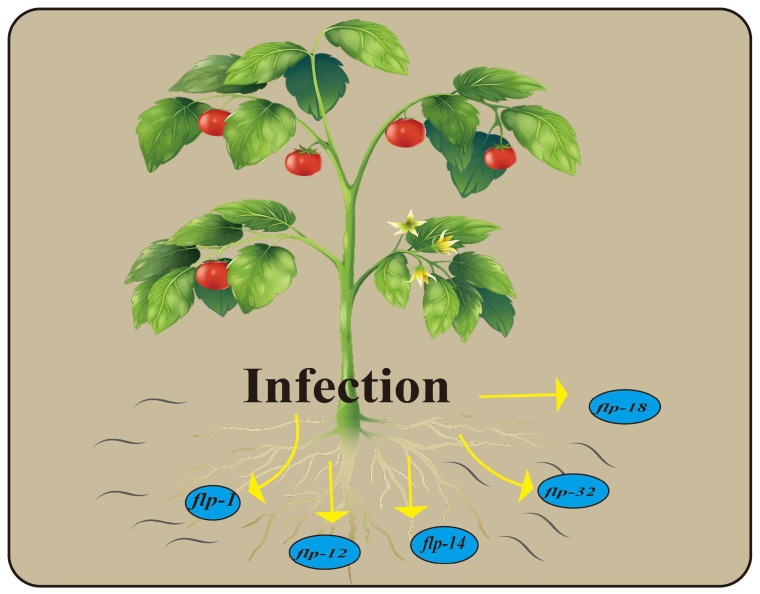
Related GPCRs of host plants with plant nematodes.

**Figure 4 ijms-26-02822-f004:**
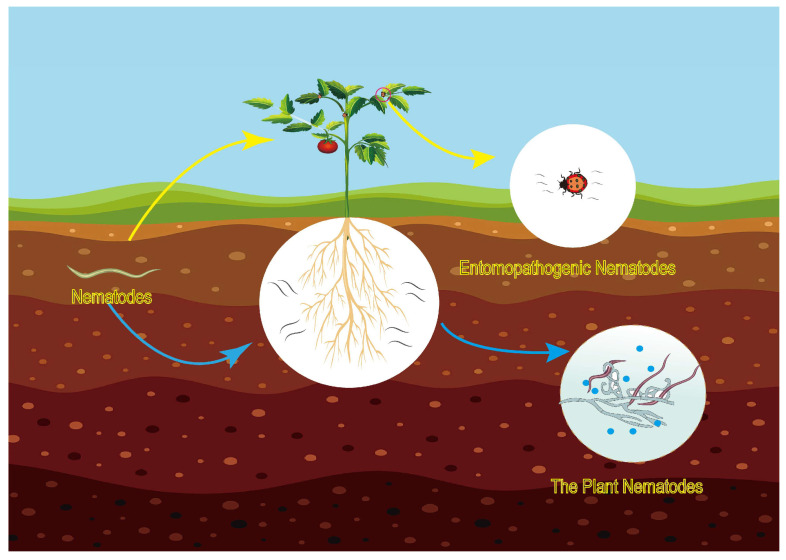
Application and control of parasitic nematodes.

**Table 1 ijms-26-02822-t001:** The interaction relationships between nematodes and other species.

Nematodes	Life Style	Interacting Species	GPCRs
*C. elegans*	saprobic	*P. aeruginosa*, *S. enterica*, *S. aureus*	*npr-8*
		*P. aeruginosa*	*octr-1*
		*P. aeruginosa*	*srbc-48*
		*E. faecalis*, *S. enterica*	*nmur-1*
		*E. coli*	*nmur-1*
		*B. nematocida* B16	*str-2*
*M. incognita*	parasitic	*N. tabacum*	*flp-14*, *flp-18*
		*O. sativa*	*flp-1*, *flp-12,*
*G. pallida*		*S. tuberosum*	*flp-32*
*M. graminicola*		*O. sativa*	*flp-18*
*B. xylophilus*		pine	*Bxy-npr-21*
*S. carpocapsae* All	parasitic	*G. mellonella*	Sc-srsx-25v, Sc-srsx-3ii, Sc-srsx-22i, and Sc-srsx-24ii, Sc-npr-23
*S. carpocapsae* UK1		*G. mellonella*	Sc-srsx-25v, Sc-srsx-3ii, Sc-srsx-22i, and Sc-srsx-24ii, Sc-npr-23, Sc-srt-62
*S. carpocapsae* All		insect	*flp-21*
*S. feltiae*		insect	*flp-3*, *flp-7*, *flp-14*, *flp-18*
*H. bacteriophora*		insect	Hba_15737 (*npr-9*)

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
