# Peer review of "GPCR Sense Communication Among Interaction Nematodes with Other Organisms"

_ijms, 2025, doi:10.3390/ijms26062822_

Round 1
Reviewer 1 Report
Comments and Suggestions for Authors
This review provides a comprehensive overview of GPCR-mediated communication in nematodes, covering interactions with bacteria, fungi, plants, and insects. In general, it is well-structured. GPCRs are well-studied in mammals, but their role in nematode interspecies interactions remains relatively underexplored. The paper addresses a research gap by consolidating information on nematode GPCRs across different ecological interactions.
- Since this is a review, it would be helpful to explicitly outline the methodology used in selecting and analyzing the literature. Please consider providing details on: The databases searched (e.g., PubMed, Web of Science) to ensure comprehensive coverage. The search strategy, including the keywords and filters applied. The inclusion and exclusion criteria were used to determine which studies were considered relevant. The screening and selection process, such as whether studies were assessed by multiple reviewers and how duplicates were handled. Including these details would enhance the transparency and reproducibility of the review.
- Given the extensive discussion of GPCR functions across different nematodes, a comparative summary table would significantly improve readability and allow for quick reference.
- While the review provides a broad and informative discussion, incorporating more recent studies (published between 2020 and 2024) would help ensure that the findings reflect the latest advancements in GPCR research.
- The potential of GPCRs as targets for pest control is an exciting and impactful research area. However, the review currently lacks direct experimental evidence demonstrating practical applications of GPCR manipulation in nematode management. To strengthen this, you may consider: Clearly differentiating between well-established findings and hypothetical applications. Discussing current limitations and challenges in translating GPCR research into field-applicable pest control methods.
Author Response
- Since this is a review, it would be helpful to explicitly outline the methodology used in selecting and analyzing the literature. Please consider providing details on: The databases searched (e.g., PubMed, Web of Science) to ensure comprehensive coverage. The search strategy, including the keywords and filters applied. The inclusion and exclusion criteria were used to determine which studies were considered relevant. The screening and selection process, such as whether studies were assessed by multiple reviewers and how duplicates were handled. Including these details would enhance the transparency and reproducibility of the review.
Re: Thank you for your positive comments and valuable suggestions. This article primarily retrieves relevant studies on the interactions between nematodes and other species using databases such as PubMed, Web of Science, ScienceDirect, and Wiley Online Library. The main keywords selected include "entomopathogenic nematodes", "C. elegans", "plant nematodes", "interaction", "communication", "nematode-trapping fungi", "plants", and "bacteria", which are connected using the conjunction "and". Literature from the past decade is considered. The inclusion criteria stipulate that the literature must focus on the direct or indirect interactions between nematodes and other species (including plants, animals, and microorganisms), primarily encompassing research validated by experiments, with a focus on English literature. Articles unrelated to the interactions between nematodes and other species, as well as non-original research papers, are excluded. The filtered literature is then intensively reviewed to assess the relevance of the research content to this article. Additionally, the quality of the filtered literature is evaluated to determine its usability. After completing the first draft, modifications and suggestions were made by experts and educators dedicated to nematode research.
- Given the extensive discussion of GPCR functions across different nematodes, a comparative summary table would significantly improve readability and allow for quick reference.
Re: Thank you for pointing this out. We have added a summary table on line 546.
- While the review provides a broad and informative discussion, incorporating more recent studies (published between 2020 and 2024) would help ensure that the findings reflect the latest advancements in GPCR research.
Re: We feel great thanks for your professional review work on our article. We have added the relevant research content from 2020 to 2024 at line 83, line 295, line 378 and line 473.
- The potential of GPCRs as targets for pest control is an exciting and impactful research area. However, the review currently lacks direct experimental evidence demonstrating practical applications of GPCR manipulation in nematode management. To strengthen this, you may consider: Clearly differentiating between well-established findings and hypothetical applications. Discussing current limitations and challenges in translating GPCR research into field-applicable pest control methods.
Re: Thank you very much for your review. We have discussed the current limitations and challenges from line 548 to line 566.

Reviewer 2 Report
Comments and Suggestions for Authors
In this paper, the authors show how interactions between nematodes and other species are possible. The authors investigate the role of GPCRs in communication between nematodes and other organisms. Some minor revision and text adjustments are required.
Critics and suggested improvements
General
- In the introduction, underline more strongly the potential applications of this study such as in the field pests control or in the pharmaceutical field
- Provide an explanation of important abbreviations in the text or add a section for abbreviations
Text
- Text (lines 54-56): give more details about the amino acids sequence such as the most important amino acids residues
- Text (lines 105-107): give examples of ligands
- Text (line 269): write what the abbreviation “ASH” means
- Text (line 269): write what the abbreviation “ASI” means
- Text (line 285): write what the abbreviation “AWC” means
- Text (line 295): write what the abbreviation “ADL” means
- Text (line 309): write what the abbreviation “AIB” means
- Text (line 397): typo in “agentss”
- Text (line 410): write the organism “ incognita” in italics
- Text (line 483): write the organism “Caenorhabditis elegans” in italics
Author Response
General
- In the introduction, underline more strongly the potential applications of this study such as in the field pests control or in the pharmaceutical field
Re: we feel great thanks for your professional review work on our article. We have described the role of GPCR in pest control on line 104-line 109, and emphasized the potential applications of this paper in pest control on line 115-line 117.
- Provide an explanation of important abbreviations in the text or add a section for abbreviations
Re: thank you very much for providing us with your valuable comments and suggestions on our research. We have added the explanations of important abbreviations in lines 576 of the manuscript.
Text
- Text (lines 54-56): give more details about the amino acids sequence such as the most important amino acids residues
Re: Thank you very much for your review. We have added the relevant content about amino acid residues on line 56-line 70.
- Text (lines 105-107): give examples of ligands
Re: thank you for pointing this out. We have added an example of the relevant ligand in line 131-line 132.
- Text (line 269): write what the abbreviation “ASH” means
Re: thank you for your reminder. We have added the relevant explanation on line 294.
- Text (line 269): write what the abbreviation “ASI” means
Re: thank you for your reminder. We have added the relevant explanation on line 295.
- Text (line 285): write what the abbreviation “AWC” means
Re: thank you for your reminder. We have added the relevant explanation on line 281.
- Text (line 295): write what the abbreviation “ADL” means
Re: thank you for your reminder. We have added the relevant explanation on line 327.
- Text (line 309): write what the abbreviation “AIB” means
Re: thank you for your reminder. We have added the relevant explanation on line 341.
- Text (line 397): typo in “agentss”
Re: We were really sorry for our careless mistakes. We have made the modification on line 436.
- Text (line 410): write the organism “ incognita” in italics
Re: We were really sorry for our careless mistakes. We have made the modification on line 450.
- Text (line 483): write the organism “Caenorhabditis elegans” in italics
Re: We were really sorry for our careless mistakes. We have made the modification on line 527.

Round 2
Reviewer 1 Report
Comments and Suggestions for Authors
All comments have been addressed